# Morphological Changes, Antibacterial Activity, and Cytotoxicity Characterization of Hydrothermally Synthesized Metal Ions-Incorporated Nanoapatites for Biomedical Application

**DOI:** 10.3390/ph15070885

**Published:** 2022-07-18

**Authors:** Ssu-Meng Huang, Shih-Ming Liu, Wen-Cheng Chen, Chia-Ling Ko, Chi-Jen Shih, Jian-Chih Chen

**Affiliations:** 1Advanced Medical Devices and Composites Laboratory, Department of Fiber and Composite Materials, Feng Chia University, Taichung City 402, Taiwan; dream161619192020@gmail.com (S.-M.H.); 0203home@gamil.com (S.-M.L.); d830191@gamil.com (J.-C.C.); 2Department of Fragrance and Cosmetic Science, College of Pharmacy, Kaohsiung Medical University, Kaohsiung 807, Taiwan; rayko1024.rb@gmail.com; 3School of Dentistry, College of Dental Medicine, Kaohsiung Medical University, Kaohsiung 807, Taiwan; cjshih@kmu.edu.tw; 4Department of Medical Research, Kaohsiung Medical University Hospital, Kaohsiung 807, Taiwan; 5Department of Orthopedics, College of Medicine, Kaohsiung Medical University, Kaohsiung 807, Taiwan; 6Department of Orthopedics, Kaohsiung Medical University Hospital, Kaohsiung 807, Taiwan

**Keywords:** hydroxyapatite, nanomaterials, antibacterial, cytotoxicity, hydrothermal synthesis, bioceramics

## Abstract

The objective of this study was to prepare hydroxyapatite (HA) with potential antibacterial activity against gram-negative and gram-positive bacteria by incorporating different atomic ratios of Cu^2+^ (0.1–1.0%), Mg^2+^ (1.0–7.0%), and Zn^2+^ (1.0–7.0%) to theoretically replace Ca^2+^ ions during the hydrothermal synthesis of grown precipitated HA nanorods. This study highlights the role of comparing different metal ions on synthetic nanoapatite in regulating the antibacterial properties and toxicity. The comparisons between infrared spectra and between diffractograms have confirmed that metal ions do not affect the formation of HA phases. The results show that after doped Cu^2+^, Mg^2+^, and Zn^2+^ ions replace Ca^2+^, the ionic radius is almost the same, but significantly smaller than that of the original Ca^2+^ ions, and the substitution effect causes the lattice distance to change, resulting in crystal structure distortion and reducing crystallinity. The reduction in the length of the nanopatites after the incorporation of Cu^2+^, Mg^2+^, and Zn^2+^ ions confirmed that the metal ions were mainly substituted during the growth of the rod-shape nanoapatite Ca^2+^ distributed along the longitudinal site. The antibacterial results show that nanoapatite containing Cu^2+^ (0.1%), Mg^2+^ (3%), and Zn^2+^ (5–7%) has obvious and higher antibacterial activity against gram-positive bacteria Staphylococcus aureus within 2 days. The antibacterial effect against the gram-negative bacillus *Escherichia coli* is not as pronounced as against *Staphylococcus aureus*. The antibacterial effect of Cu^2+^ substituted Ca^2+^ with an atomic ratio of 0.1~1.0% is even better than that of Mg^2+^- and Zn^2+^- doped with 1~7% groups. In terms of cytotoxicity, nanoapatites with Cu^2+^ (~0.2%) exhibit cytotoxicity, whereas Mg^2+^- (1–5%) and Zn^2+^- (~1%) doped nanoapatites are biocompatible at low concentrations but become cytotoxic as ionic concentration increases. The results show that the hydrothermally synthesized nanoapatite combined with Cu^2+^ (0.2%), Mg^2+^ (3%), and Zn^2+^ (3%) exhibits low toxicity and high antibacterial activity, which provides a good prospect for bypassing antibiotics for future biomedical applications.

## 1. Introduction

Antibiotics are the most common drugs currently used to fight bacterial infections to prevent the emergence of biofilms. However, several types of bacteria can develop antibiotic resistance due to a combinatorial drive of bacteria exposed to antibiotics [1]. The current treatment of bacterial biofilms is complex, expensive and inefficient, leading to an urgent need for effective alternative methods [2,3]. One promising solution is functional dry powders, which are attractive and offer several advantages over other formulations. For example, when dry powder is used, therapeutic doses can be delivered to more targeted locations, usually at the site of infection [4]. Hydroxyapatite (HA), a member of the apatite family that can be synthesized at the nanoscale using chemical precursors, is a known biocompatible material that is widely used for replacement and regeneration of bone materials. HA is commonly used as dry powder with the molecular formula Ca_10_(PO_4_)_6_(OH)_2_ and is a calcium phosphate-based bioceramic [5]. Apatite has excellent biocompatibility and osteoconductivity, similar to inorganic compounds in human bones and teeth [6]. Therefore, it is often used as a bone graft substitute in biomedical clinics [7]. Different forms of apatite with different sizes and shapes (e.g., flakes, spheres, needles, and rods) have been fabricated using wet and dry synthesis strategies [8,9,10,11,12,13,14,15,16,17,18,19,20,21,22]. HA can be synthesized by a precipitation method [8,9], sol-gel method [10,11], hydrothermal method [8,9,10] and microwave-assisted synthesis method [15,16], or it can be directly extracted from biological resources [17,18,19]. Reactions during hydrothermal wet precipitation can be effectively controlled by adjusting the reaction medium, pH value, and concentration at different temperatures, amounts of pressure, and reaction times [23,24,25]. The reaction conditions are relatively mild, and the advantage is that the stoichiometric ratio, structural morphology, uniformity, and apatite size can be controlled.

Although apatite offers benefits as a biomaterial implant, it can affect the proliferation rate of osteoblasts and the expression of cytokine genes [26,27]. However, pure apatite dry powder lacks an antibacterial mechanism. When an infection occurs after surgery, pathogenic bacteria rapidly aggregate to form a biofilm, which strengthens the bacteria’s defense against the host and produces drug resistance. Appropriate antibiotic prophylaxis reduces the risk of postoperative wound infection, but antibiotics are becoming increasingly ineffective as resistance spreads globally [27]. As a result, treating infections and preventing deaths becomes challenging. This study attempted to use metal ions in preparing antibacterial nanoapatites to address problems caused by drug-resistant “super bacteria” [27,28].

A nanoapatite crystal has a hexagonal structure with two distinct charged sites (positively charged Ca longitudinal and negatively charged P transverse sites). Normally added positively charged metal ions enter the lattice to displace Ca^2+^ sites, but some bind to negatively charged P sites [28,29,30]. Nanoapatites prepared by the co-precipitation of metal ions with Ca^2+^ can target bacterial cells; they penetrate and damage the cell membrane, promote protein alkylation, and cause oxidative stress, thus resulting in genetic material damage and death [31]. Adding appropriate amounts of metal ions (e.g., Cu^2+^, Mg^2+^, Zn^2+^, Ag^+^, Sr^2+^, Li^+^, and Si^4+^) to the HA lattice does not affect the structural configuration and biocompatibility and can effectively change the lattice, crystallinity, solubility, bacteriostatic, and angiogenesis of nanoapatites [32,33,34,35,36,37]. Cu^2+^, Mg^2+^, and Zn^2+^ metal ions with nearly the same anion radius were investigated as targets for the hydrothermal synthesis of doped nanoapatites. The metal ion Cu^2+^ is an essential trace element for the human body and exerts an antibacterial effect while promoting bone formation, inhibiting osteoporosis, and promoting human endothelial cell proliferation [38,39,40]. Approximately 60% of Mg^2+^ in the human body is stored in the bones, and Mg^2+^ regulates osteoblast or osteoclast activities [41]. Thus, Mg^2+^ deficiency can cause skeletal growth arrest and osteopenia. Mg^2+^ induces endothelial cells to produce nitric oxide, leading to migration and growth and thereby accelerating the formation of new blood vessels [41,42,43]; Zn^2+^ increases the activities of osteoblasts, activates bone formation, promotes the synthesis of collagen, and inhibits the bone resorption of osteoclasts [44,45,46].

Here, we proposed the co-precipitation of Cu^2+^, Mg^2+^, and Zn^2+^ with Ca^2+^ for the preparation of nanoapatites with antibacterial properties through hydrothermal wet synthesis and compared their antibacterial properties and biocompatibility. The composition, phase identification, morphological changes, antibacterial activity and cytotoxicity of synthesized nanoapatite doped with Cu^2+^, Mg^2+^, and Zn^2+^ ions were investigated.

## 2. Results and Discussion

### 2.1. FT-IR Spectroscopic Analysis

The characteristic apatite absorption bands are shown in Figure 1. Each group showed the librational vibration (ν_L_) and asymmetric stretching absorption (ν_3_) of hydroxyl (OH^−^) groups observed at wavenumbers of 632 and 3568 cm^−1^, respectively. The apatite PO_4_^3–^ absorption bands at wavenumbers 563 and 603 cm^−1^ were observed and attributed to the bending vibration of O−P−O (triply degenerate δ4 bending in PO_4_^3−^). The symmetric stretching band (ν_1_ mode in PO_4_^3−^) at 964 cm^−1^ and asymmetric stretching bands (ν_3_ mode in PO_4_^3−^) at 1034 and 1097 cm^−1^ were detected [47,48,49,50]. In addition, the NH_4_^+^ functional group was found at 1385 cm^−1^, and the C–H symmetrical and asymmetric stretching vibration bands were observed at 2854 and 2924 cm^−1^, respectively, which were attributed to the diammonium hydrogen phosphate ((NH_4_)_2_HPO_4_) and residual functional groups of sodium citrate (Na_3_C_6_H_5_O_7_). The C=O stretching vibration bands of CO_2_ were observed at 1448 and 1635 cm^−1^ [51].

### 2.2. XRD Analysis

To elucidate the phase structure and crystallinity produced by the hydrothermal reaction, the XRD patterns of nanoapatites with different Cu^2+^, Mg^2+^, and Zn^2+^ ion doping levels are illustrated in Figure 2 The patterns were compared with those in the JCPDS database. The diffraction planes (002), (211), (112), and (300) were the main characteristic peaks of HA (JCPD 73−0294, JCPD 74−0565, and JCPD 74−0566). No other characteristic compounds of the diffraction planes containing Cu, Mg, and Zn were detected. The measured increase in the full width at half maximum (FHMW) of most planes observed in Table 1 reflects a decrease in crystallinity as doping metal ions increased in the nanoapatites. This decrease was attributed to the uneven replacement of the apatite bonding sites of Ca^2+^ (1.14 Å) by Cu^2+^ (0.87 Å), Mg^2+^ (0.86 Å), and Zn^2+^ (0.88 Å) ions, resulting in lattice distortion in nanoapatite crystalline structures [52,53,54].

### 2.3. TEM Analysis

Figure 3 shows the TEM images and SAED analysis results of the original nanoapatite rod and nanoapatite rods doped with different ions (Cu^2+^, Mg^2+^, or Zn^2+^). Each group was in the form of rod-shaped nanoapatites. In SAED analysis, the diffraction patterns of the characterized (002), (211), and (112) planes and the lattice distances in HRTEM were observed, and HA was observed to correspond to 0.34 and 0.28 nm of the (002) and (300) planes, respectively. This result was verified with the XRD patterns. The doped and undoped rod-shaped nanoapatites had average lengths of 31–49 nm, average widths of 6–12 nm, and average aspect length-to-width ratios of 2–5. No significant differences in the plane ratios between length (002) and width (300) in the lattice calculated from the FWHM value of the XRD pattern were found, but the plane ratio values of Cu^2+^, Mg^2+^, and Zn^2+^ were greater than the value of the Ca^2+^-only control group (Table 2). Except in 0.1Cu−nHA, 1Mg−nHA, and 1Zn−nHA groups with low amounts of doped ions, the lengths of the rod-shaped nanoapatites decreased with increasing amounts of doped ions (*p* < 0.05).

According to Shannon’s ionic radii in a previous study [55], which reported the difference in coordination number and high and low spin states of ions, Ca^2+^ (1.14 Å) at the longitudinal sites of a nanoapatite were mainly replaced by Cu^2+^ (0.87 Å), Mg^2+^ (0.86 Å), and Zn^2+^ (0.88 Å) with small ionic radii. The average transverse width of 3Mg−nHA (6.80 ± 1.74 nm) was smaller than that of nHA (10.44 ± 2.38 nm; *p* < 0.05) and more elongated, but the average width of 7Zn−nHA (12.05 ± 2.62 nm) was larger than that of nHA (*p* < 0.05), making it short and wide.

### 2.4. Antibacterial Activity

The results of the quantitative analysis of the antibacterial effect of nanoapatite hydrothermally synthesized with different concentrations of Cu^2+^, Mg^2+^, and Zn^2+^ ions on *S. aureus* are shown in Figure 4. In the hydrothermal synthesis of nanoapatites, the test groups with Cu^2+^, Mg^2+^, and Zn^2+^ ions exhibited better bacteriostatic ability than the control nHA group without metal ion addition, especially on day 1 (*p* < 0.05). As for the individual antibacterial strength of each ionic group, the antibacterial ability of the 0.1Cu−nHA group at a low concentration was comparable to that of 1Cu−nHA at a high concentration until day 3 in terms of strength (Figure 4a). As shown in Figure 4b, the slender nanoapatite of the 3Mg−nHA group showed a higher inhibitory effect in the group containing Mg^2+^. As for the group with Zn^2+^, as the Zn^2+^ ionic addition amount increased, the bacteriostatic ability was enhanced (Figure 4c).

Figure 5 shows the quantitative analysis results for the antibacterial effects of the tested nanoapatite groups synthesized with Cu^2+^, Mg^2+^, and Zn^2+^ ions on *E. coli* for 1–4 days, and the concentrations of the groups were compared with the concentration of the control nHA group. Each test group had an antibacterial effect on *E. coli* on the first day of the analysis. Regarding the antibacterial effect of each ion, although the Cu^2+^ ability of 0.3Cu−nHA and 1Cu−nHA groups against *E. coli* increased with Cu^2+^ concentration, the antibacterial ability of 0.1Cu−nHA with a lower concentration of Cu^2+^ was maintained for 2 days (Figure 5a). The test groups of 1Mg−nHA and 3Mg−nHA exhibited antibacterial ability against *E. coli* in 2 days of testing (Figure 5b). Except in the 7Zn−nHA group, the antibacterial ability of Zn^2+^ was still statistically significant (*p* < 0.05) compared with that in the nHA control group on day 2 (Figure 5b), but the difference in antibacterial effect was small. The bacteriostatic ability of the test group with Cu^2+^, Mg^2+^, and Zn^2+^ ions for *S. aureus* was better than that for *E. coli*.

The cell wall structure of *E. coli* is more complex than that of *S. aureus* [56], and thus Cu^2+^, Mg^2+^, and Zn^2+^ ions can barely disrupt the cell wall of *E. coli* and hardly exert a bacteriostatic effect [57,58]. In addition, nanoapatites with low concentrations of Cu^2+^ ions exhibited better bacteriostatic effects than those with Mg^2+^ and Zn^2+^ [59]. The particle size of Cu^2+^, Mg^2+^ and Zn^2+^ ions (~0.86–0.88 Å) is much smaller than that of Ca^2+^ (1.14 Å), which causes the possibility of free entry and exit of bacterial cells, destroys cell wall synthesis, and reduces the protective effect of bacteria. For example, literature suggests that Cu^2+^ ions can bind to thiol bonds in the cytoplasm, disrupting bacterial metabolism and differentiation [60]. Hence, the combined effects of the aspect ratios and surface charges of nanoapatites on the bacteriostatic ability of bacteria should be considered.

### 2.5. Cytotoxicity

Figure 6 shows the quantitative and qualitative analysis results of cytotoxicity in L929 cells incubated with nanoapatites with different metal ions for 1 day. Extracts that result in a reduction of control activity below 70% are considered cytotoxic as described in the standard ISO 10993-5 [61]. Figure 6a shows that the cell viability of HDPE was the same as that in the control nHA group, indicating that the groups were effective during the sterilization process. Meanwhile, the control nHA group without Cu^2+^, Mg^2+^, and Zn^2+^ showed no toxicity. The nanoapatite groups with Cu^2+^ showed cytotoxicity, whereas nanoapatites with Mg^2+^ and Zn^2+^ and the 7Mg−nHA, 3Zn−nHA, 5Zn−nHA, and 7Zn−nHA groups still showed cytotoxicity [62]. The cell morphology in Figure 6b was qualitatively analyzed, and the results of each group showed the same trend as the quantitative analysis results of cytotoxicity.

The preparation of metal ion-incorporated nanoapatites with facile synthetic techniques is a significant topic. These nanoapatites inhibit drug resistance and biofilm formation, especially during open wound repair. Functional nanoapatites with anitbacterial ability are clinically useful in preventing the proliferation of bacteria with resistance genes and biofilm-producing bacteria, such as *S. aureus* resistant to asmethicillin and vancomycin, in implants and related in vitro and in vivo medical devices [31]. The World Health Organization has highlighted the importance of these bacteria, listing them as strains that require the most in-depth microbial research. It mentions all key groups that pose a particular threat to hospitals, nursing homes, and patients in need of medical equipment and can cause serious and often fatal infections, such as bloodstream infections and pneumonia. Nanoparticles exhibit broad antimicrobial activity against gram-positive bacteria (*Enterococcus*, *Staphylococcus*, and *Streptococcus*) and gram-negative bacteria (*E. coli* and *Pseudomonas*). However, our study and some other studies have shown that nanoapatites are more active against gram-positive bacteria than against gram-negative bacteria (Figure 4 and Figure 5). This difference may be related to the difference in cell wall structure between the bacterial types [56,57,58].

Arul et al. [63] used a microwave strategy to rapidly synthesize Mg^2+^-incorporated apatite nanorods. They found that the incorporation of Mg^2+^ ions did not alter the apatite phase, but significantly reduced the crystallinity and particle size by 48% and 32%. Their results were consistent with the results obtained from the 3Mg-nHA test group (Figure 3, second image of middle row and Table 2), which exhibited biocompatibility and antibacterial activity (Figure 4b and Figure 5b). In the agglomerated apatite shown in the TEM (Figure 3, images in the middle row), the crystallinity and particle sizes of the nanoapatites decreased with increasing Mg^2+^ concentration.

Shanmugam and Gopal [39] demonstrated that a small amount of copper ions can induce high activity or antibacterial activity in living cells, HA can be replaced by Cu^2+^ ions (Ca_10−x_Cu_x_(PO_4_)_6_(OH)_2_ (x = 0.05–2.0), and the antibacterial ratio against *S. aureus* is 0.2. Their results were confirmed in the present study (Figure 4a and Figure 5a), although their Cu^2+^-substituted nanoapatite was synthesized by precipitation and sintering at 700 °C.

Zn^2+^ ion substitution in nanoapatites has been a subject of particular interest because it is present in all biological tissues and plays diverse roles in many biological functions [64]. Bones and teeth contain large amounts of Zn^2+^, and its absorption or release is strongly mediated by bone depots. Tang et al. found that Zn^2+^ is substituted in the apatite structure to form an apatite-like phase and then forms an amorphous structure with parascholzite (CaZn_2_(PO_4_)_2_ ∙2H_2_O), which is the most favorable condition for the occurrence of Zn^2+^ in tetrahedral coordination and Ca2 sites with obvious local structure distortion. In the coordination of Ca1 and P sites with Zn, all higher shells move to longer and shorter distances and lead to elongated crystallites. These results contradict our results in Figure 3, images in the bottom row. Therefore, the fitting to Zn^2+^ at the Ca2 site leads to the overall shrinkage of the local structure, most of the higher shells move toward Zn, and thus the high degree of Zn^2+^ incorporation resulted in the shorter morphology of the nanoapatite (Table 2).

Nanoapatite constitutes the major inorganic components of bones and teeth and can thus improve mechanical properties combined with nanotopographic features that mimic natural bone nanostructures. In this study, we discovered nanoapatite with metal ions, which can be functionalized because of its biocompatibility and effects that inhibit bacterial action. Therefore, the synthesized apatite exhibited antibacterial activity when co-precipitated with Cu^2+^, Mg^2+^, and Zn^2+^, especially in the 0.2Cu−nHA nanoapatite, which showed the highest antibacterial activity and best compatibility in the Cu-incorporated group (Figure 4, Figure 5 and Figure 6), although still exhibiting cytotoxicity under ISO10993-5 test conditions. The Mg^2+^ group had the best biocompatibility. The antibacterial effect of nanoapatite incorporated with Zn^2+^ was poor and showed cytotoxicity at high Zn^2+^ concentrations [59]. Most of the summarized differences in the antimicrobial properties of Zn^2+^ can be attributed to the concentrations studied. Therefore, the relative suggestion of the antimicrobial effect of Cu^2+^ over Mg^2+^ and Zn^2+^ in this study is also concentration-dependent and takes into account the biocompatibility. Another possibility that is different from other literature is that the nanoapatite dry powder does not release metal ions but interacts with the bacterial cell surface, thereby changing the cell permeability, or forming an impermeable layer around the cell, preventing essential solutes from entering the cell.

## 3. Materials and Methods

### 3.1. Raw Materials

The materials used in this study were calcium nitrate (Ca(NO_3_)_2_ ∙4H_2_O, purity > 98.0%, KATAYAMA CHEMICAL Co., Ltd., Osaka, Japan), diammonium phosphate ((NH_4_)_2_HPO_4_, purity > 98.0%, HSE PURE CHEMICALS, Ahmedabad, India), sodium citrate (Na_3_C_6_H_5_O_7_, purity > 99.0%, PANREAC, Barcelona, Spain), copper (II) nitrate (Cu(NO_3_)_2_ ∙3H_2_O, purity > 99.0%, PANREAC, Barcelona, Spain), magnesium nitrate (Mg(NO_3_)_2_ ∙6H_2_O, purity > 98.0%, PANREAC, Barcelona, Spain), zinc nitrate (Zn(NO_3_)_2_ ∙6H_2_O, purity > 98.0%, PANREAC, Barcelona, Spain), acetic acid (CH_3_COOH, purity > 99.0%, PANREAC, Barcelona, Spain), and sodium hydroxide (NaOH, purity > 99.0%, SHIMAKYU CHEMICAL Co., Ltd., Osaka, Japan).

### 3.2. Preparation of Nano-Hydroxyapatite with Metal Ions (Cu^2+^, Mg^2+^, and Zn^2+^)

The hydrothermal synthesis of nanoapatite (nHA), Cu^2+^, Mg^2+^ and Zn^2+^ ion-doped nanoapatite followed the previously reported research procedure [65]. The concentrations of Cu^2+^, Mg^2+^, and Zn^2+^ ions were set at molar levels different from the total molar content of metal cations, as shown in Table 3. Briefly, the prepared cation solutions with different ratios of Ca(NO_3_)_2_ ∙4H_2_O and different amounts of Cu^2+^, Mg^2+^, and Zn^2+^ ions (as indicated in Table 3) were dissolved and mixed into 90 mL of double-distilled water (ddH_2_O). The pH of the resulting solution was adjusted to 7. Then, 0.713 g of (NH_4_)_2_HPO_4_ and 2.647 g of Na_3_C_6_H_5_O_7_ were stirred with 70 mL of ddH_2_O for 30 min to form an anion solution. The aforementioned two cation and anion solutions were mixed and stirred for an additional 30 min. The nanoapatites were synthesized through hydrothermal synthetic reaction at 180 °C and 220 psi (l b/in^2^) for 12 h. After the reaction, the nanoapatites were washed three times with ethanol and ddH_2_O at 3000 rpm for 5 min each time and then dried at 60 °C for 12 h. The yield of the resulting precipitate in this study was about 30% (*w/w*).

### 3.3. Characterization of Nanoapatites through Fourier Transform Infrared (FTIR) Spectroscopy, X-ray Diffraction (XRD) Analysis, and Transmission Electron Microscopy (TEM) Morphological Measurements

The functional groups of the nanoapatites were analyzed using a Fourier transform infrared (FTIR) spectroscopy (Nicolet 6700, Thermo Fisher Scientific, Waltham, MA, USA) to confirm the structural composition [66].

The crystalline phases, crystallinity, and preferred orientation changes in the nanoapatites were investigated through X-ray diffraction (XRD) pattern analysis. Diffraction analysis was performed using an X-ray diffractometer (D2 Phaser, Bruker, Billerica, MA, USA), and the diffraction conditions were as follows: Kα diffraction using a Ni-filtered Cu target at a voltage of 30 kV, current of 20 mA, and scan speed of 2°/min. The scanning range of 2*θ* was 20–60°, and the phases were identified by comparing the diffraction patterns with the patterns in the Joint Committee on Powder Diffraction Standards (JCPDS).

The sample was mixed with 99.8% alcohol (Avantor, Inc., Radnor, PA, USA) and dispersed uniformly with an ultrasonic oscillator (DC400H, DELTA ULTRASONIC Co., Ltd., New Taipei, Taiwan). The dispersion solution was dropped onto a copper mesh (TED PELLA, INC., Redding, CA, USA). The morphology of HA was analyzed using transmission electron microscopy (TEM; JEM-2100F, JEOL, Tokyo, Japan) at 200 kV and bright-field images. The microstructure of HA was observed using a selected area electron diffraction (SAED) system with an aperture size of 160 nm, and the crystal structure was analyzed through high-resolution transmission electron microscopy (HRTEM). Digital Micrograph (Gatan) software was used to measure the diffraction plane distance (d). Image-J program (software version 1.53e) was used to analyze the TEM images of each group of samples and calculate the lengths, diameters, and aspect ratios of the nanoapatites.

### 3.4. Antibacterial Abilities

*Staphylococcus aureus* (ATCC No. 25923) and *Escherichia coli* (ATCC No. 10798) were cultured in Luria–Bertani broth [67,68,69]. The bacterial suspension was diluted to achieve an optical density at 595 nm (OD_595_) of 0.2 (equivalent to ~1.0 × 10^7^ cells/mL on average) and this value was confirmed with an enzyme-linked immunosorbent assay (ELISA) reader (EZ Read-400, Biochrom, Holliston, MA, USA). The bacterial suspensions of *S. aureus* and *E. coli* were subsequently diluted to achieve an OD_595_ value of 0.2. For quantitative antibacterial testing, 0.03 g of a sample was immersed in 2 mL of bacterial suspension and incubated at 37 °C for 1–4 days. Approximately 100 µL of the bacterial solution was collected from each group, and its OD_595_ value was obtained with the ELISA reader (*n* = 3).

### 3.5. In Vitro Cytotoxicity Tests

The L929 cell line from newborn mouse fibroblasts was provided by the National Institute of Health in Taiwan and used for the cytotoxicity tests. The testing procedures were conducted by ISO 10993-5:2009. L929 cells were grown with media in an incubator at 37 °C and 5% CO_2_ and subcultured when the cell concentration was between 0.8 × 10^6^ and 1.0 × 10^6^ cells/mL. The medium used was a minimal essential medium alpha medium (Gibco, Thermo Fisher Scientific Inc., Waltham, MA, USA) containing 10% horse serum and was changed every 2 days of culture.

The samples prepared for cell culture were sterilized by autoclaving at 121 °C and 1.05 kg/cm^2^ (15–20 psi; TOMIN, TM-328, Taipei, Taiwan). The extraction solution was prepared by placing the sterilized nanoapatites, a negative control group of high-density polyethylene (HDPE) and a positive control group of 15 vol.% dimethyl sulfoxide (DMSO; Sigma-Aldrich, St Louis, MO, USA) in the culture medium. Owing to the hydrophilicity of the nanoapatites [70], the extraction weight ratio (g/mL) of the sample to the medium was set at 1:5, and the samples were placed in a 37 °C incubator for 24 h, and the extracted medium was transferred for cytotoxicity testing.

For the quantitative cytotoxicity test, 100 μL of suspension with an L929 cell concentration of 1 × 10^4^ cells was transferred to a 96-well microliter plate and cultured at 37 °C for 24 h in an incubator. The medium was removed, and then 100 μL of the sample extract was added. L929 cells were incubated in a 37 °C incubator for 24 h and then mixed with 50 μL of a XTT cell proliferation assay kit (Biological Industries, Kibbutz Beit Haemek, Israel) for a 4 h extension reaction before treatment with another ELISA reader (SPECTROstar Nano, BMG LABTECH, Offenburg, Germany). The measured OD_490_ absorbance was proportional to cell viability.

In the qualitative cytotoxicity test, the preparation of the extracts and the culture medium of the control group were the same as those in the quantitative test. Approximately 1000 μL of cell suspension was collected, and L929 cells were transplanted into a 48-well microliter plate at a cell concentration of 1 × 10^5^ cells/well. The original medium was cultured in an incubator at 37 °C and 5% CO_2_ for 1 day, and then the medium was removed. Sample extract (100 μL) was added and cultured for 24 h, and cell morphology was observed under an inverted microscope (IVM-3AFL, SAGE VISION Co., Ltd., New Taipei City, Taiwan).

### 3.6. Statistical Analysis

Analysis of variance (ANOVA) was performed using IBM SPSS Statistics version 20 (SPSS Inc., Chicago, IL, USA) for the measurement of average longitudinal length, transverse width, length-to-width aspect ratio and antibacterial activity. ANOVA was used in determining whether differences among the means of multiple groups were significant. The estimates of two different variables were used in comparing the differences.

## 4. Conclusions

In summary, almost all nanoapatites with metal ions (Cu^2+^, Mg^2+^, and Zn^2+^) exhibited initial antibacterial activity, the magnitude of which was Cu^2+^, Mg^2+^, and Zn^2+^ in descending order. After the addition of metal ions, the main characteristic peaks were slightly shifted due to the smaller size of the HA crystal structure. In particular, the length shrinkage of the nanorods. Cu^2+^, Mg^2+^, and Zn^2+^ replaced Ca^2+^, resulting in changes in the lattice structure, deformation of the crystal structure, and a decrease in crystallinity. In this formulation, Cu^2+^, Mg^2+^, and Zn^2+^ ions replaced Ca^2+^ in the apatite mainly at the longitudinal positions of the nanorods. Antibacterial test results showed that adding Cu^2+^ ions with an atomic ratio of 0.1% was more effective than doping Mg^2+^ (3%) and Zn^2+^ (5–7%) ions, although they both showed gram-positive antibacterial activity. Nanoapatite with all Cu^2+^ ion concentrations was toxic, but 0.2% Cu^2+^ showed relatively little cytotoxicity in the group. After adding Mg^2+^ ions to the nanoapatite, only the 7% Mg^2+^ group with the highest doping amount showed slight cytotoxicity and was the test group with the highest biocompatibility. For Zn^2+^ ionic groups that exhibit moderate cytotoxicity, doping at 1% of the Zn^2+^ groups is expected to be biocompatible, while 3% of the Zn^2+^ groups have little cytotoxicity. Therefore, in terms of biocompatibility and antibacterial synergistic function of nanoapatite, the best group is 3% Mg^2+^-doped nanorods. Future studies should consider expanding the testing range; for example, if the 0.2% Cu^2+^ doped nanorods with the strongest antibacterial activity in this study were incorporated into other biocompatible medical devices, it might reduce cytotoxicity but maintain bacteriostatic properties.

## Figures and Tables

**Figure 1 pharmaceuticals-15-00885-f001:**
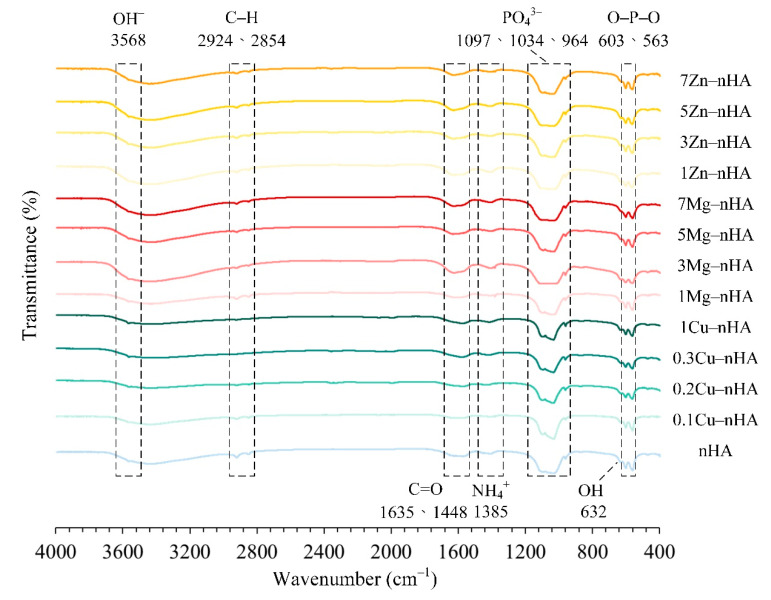
Fourier transform infrared spectra of rod-shaped nanoapatites synthesized with and without Cu^2+^, Mg^2+^, and Zn^2+^ at different ion concentrations.

**Figure 2 pharmaceuticals-15-00885-f002:**
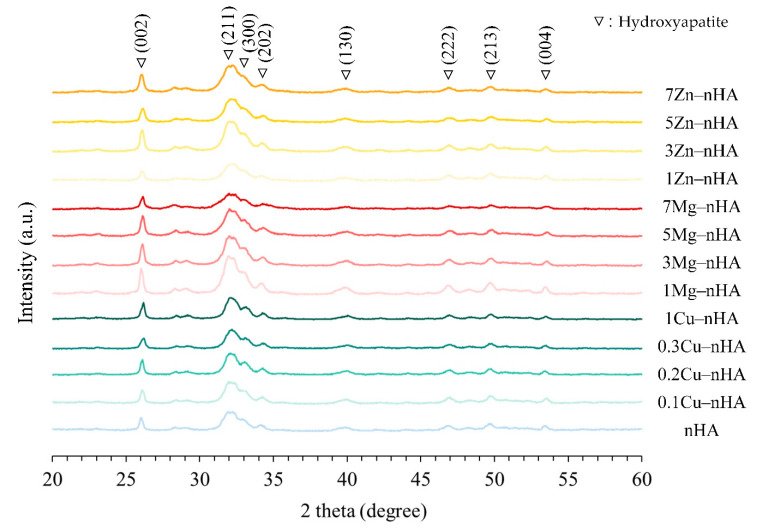
X-ray diffraction patterns of the rod-shaped nanoapatites synthesized with and without Cu^2+^, Mg^2+^, and Zn^2+^ at different ion concentrations.

**Figure 3 pharmaceuticals-15-00885-f003:**
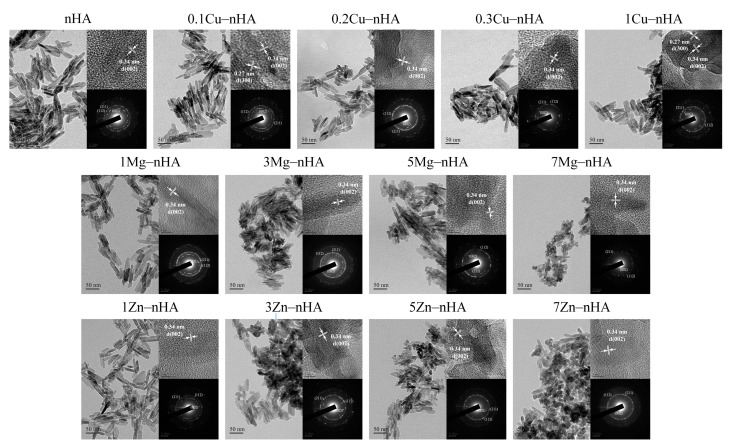
High-resolution transmission electron microscopy images, lattice images of the (002) facet of apatite were identified, and selected area electron diffraction analysis of rod-shape nanoapatites hydrothermal synthesized with different ionic co-precipitation concentrations in the absence (top left) and presence of Cu^2+^ (top row), Mg^2+^ (middle row), and Zn^2+^ (bottom row).

**Figure 4 pharmaceuticals-15-00885-f004:**
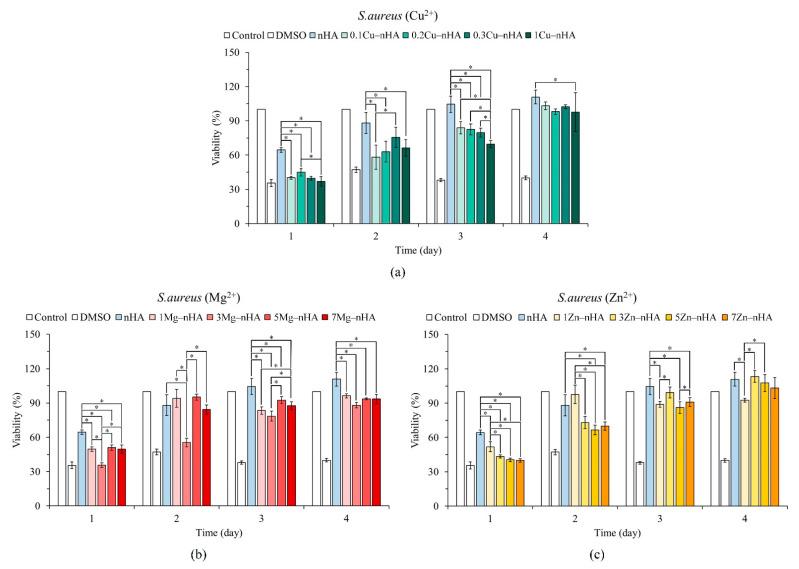
Relative quantitative antibacterial ability against gram-positive *S. aureus* in the presence of different ionic concentrations of Cu^2+^ (**a**), Mg^2+^ (**b**), and Zn^2+^ (**c**), the rod-shaped nanoapatites with different ion concentrations (*n* = 3; * indicates significantly different *p* < 0.05).

**Figure 5 pharmaceuticals-15-00885-f005:**
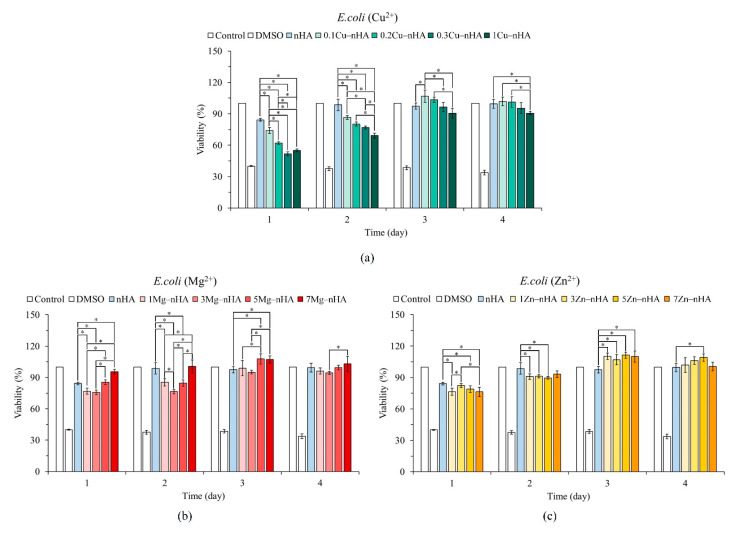
Relative quantitative antibacterial ability against gram-negative *E. coli* in the presence of different ionic concentrations of Cu^2+^ (**a**), Mg^2+^ (**b**), and Zn^2+^ (**c**), the rod-shaped nanoapatites with different ion concentrations (*n* = 3; * *p* < 0.05).

**Figure 6 pharmaceuticals-15-00885-f006:**
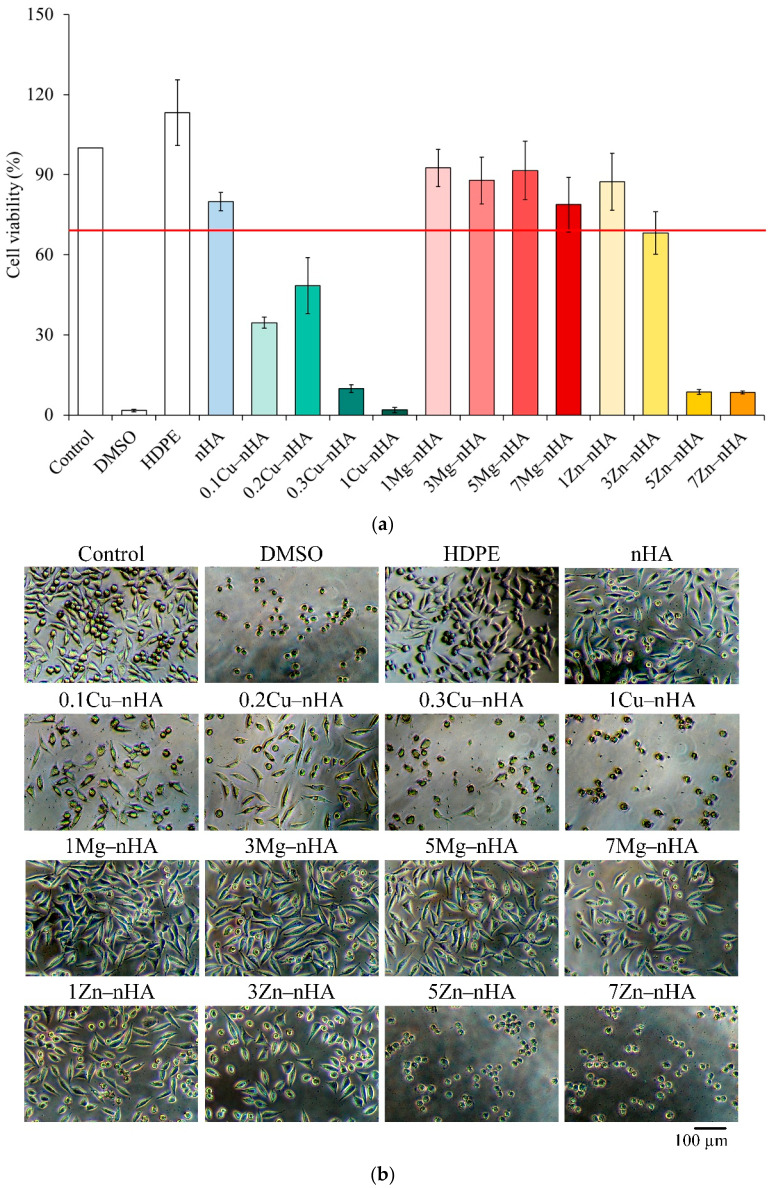
(**a**) Quantitative results of cytotoxicity (*n* = 6) after culturing L929 cells with nanoapatite doped with different metallic ions for 1 day; (**b**) qualitative analysis of cytotoxicity.

**Table 1 pharmaceuticals-15-00885-t001:** Calculation of full width at half maximum values from X-ray diffraction patterns of rod-shaped nanoapatites synthesized with different concentrations of Cu^2+^, Mg^2+^, and Zn^2+^ ions.

Samples	(002)	(211)	(300)	(202)	(130)	(222)	(213)	(004)
nHA/control	0.319	1.074	0.429	0.368	0.792	0.547	0.431	0.340
0.1Cu−nHA	0.321	1.048	0.336	0.400	0.826	0.535	0.443	0.345
0.2Cu−nHA	0.304	0.834	0.522	0.384	0.724	0.533	0.424	0.333
0.3Cu−nHA	0.378	0.861	0.464	0.347	0.696	0.577	0.425	0.378
1Cu−nHA	0.324	0.818	0.538	0.367	0.705	0.527	0.390	0.351
1Mg−nHA	0.318	1.042	0.362	0.297	0.812	0.542	0.435	0.332
3Mg−nHA	0.322	0.924	0.485	0.387	0.828	0.529	0.430	0.342
5Mg−nHA	0.345	0.956	0.442	0.442	0.827	0.518	0.410	0.384
7Mg−nHA	0.356	0.855	0.579	0.597	0.777	0.597	0.434	0.477
1Zn−nHA	0.349	0.934	0.337	0.337	0.851	0.534	0.517	0.393
3Zn−nHA	0.338	1.053	0.350	0.269	0.836	0.549	0.464	0.332
5Zn−nHA	0.374	1.082	0.334	0.361	0.866	0.621	0.510	0.361
7Zn−nHA	0.369	1.128	0.318	0.401	0.884	0.562	0.472	0.395

**Table 2 pharmaceuticals-15-00885-t002:** Average longitudinal length, transverse width, length-to-width aspect ratio (*n* = 50) of rod-shaped nanoapatites and plane ratio of (002) to (300) FWHM values after doping in different Cu^2+^, Mg^2+^, and Zn^2+^ ionic reaction concentrations.

Samples	Longitudinal Length (nm)	Transverse Width (nm)	Length-to-Width Aspect Ratio	Plane Ratio of (002) to (300) FWHM Values
nHA/control	48.76 ± 10.50	10.44 ± 2.38	4.85 ± 1.39	0.74
0.1Cu−nHA	47.07 ± 7.33	11.80 ± 2.67	4.15 ± 0.97	0.96
0.2Cu−nHA	42.65 ± 10.87	10.99 ± 2.17	3.92 ± 0.81	0.58
0.3Cu−nHA	42.73 ± 10.74	10.38 ± 2.82	4.43 ± 1.72	0.81
1Cu−nHA	41.66 ± 9.75	11.91 ± 2.77	3.58 ± 0.82 *	0.60
1Mg−nHA	46.16 ± 9.23 *	10.32 ± 2.15	4.62 ± 1.19	0.88
3Mg−nHA	37.03 ± 7.60	6.80 ± 1.74 *	5.69 ± 1.52 *	0.66
5Mg−nHA	36.48 ± 8.49	9.62 ± 2.46	3.94 ± 1.06	0.78
7Mg−nHA	34.99 ± 9.65	9.06 ± 1.95	3.96 ± 1.12	0.61
1Zn−nHA	45.69 ± 8.54 *	12.15 ± 2.62	3.91 ± 1.04	1.04
3Zn−nHA	39.37 ± 8.33 *	10.04 ± 2.34 *	4.09 ± 1.18	0.97
5Zn−nHA	31.92 ± 5.21	9.30 ± 2.14 *	3.57 ± 0.86	1.12
7Zn−nHA	30.53 ± 4.80	12.05 ± 2.62	2.62 ± 0.58 *	1.16

* *p* < 0.05.

**Table 3 pharmaceuticals-15-00885-t003:** Nomenclature of nanoapatite groups with metal ions at varying content during hydrothermal synthesis.

Abbreviated Names	Ca(NO_3_)_2_•4H_2_O (mmol)	Contents of Metal Ion in Morality	[M/(Ca + M)] × 100% ^a^	(NH_4_)_2_HPO_4_ (mmol)	(Ca + M)/P (Atomic Ratio)
**nHA/control**	9.000	−	−	5.400	1.670
**Cu**(**NO_3_**)**_2_•3H_2_O** (**mmol**)					
**0.1Cu–nHA**	8.991	0.009	0.1	5.400	1.670
**0.2Cu−nHA**	8.982	0.018	0.2	5.400	1.670
**0.3Cu−nHA**	8.973	0.027	0.3	5.400	1.670
**1Cu−nHA**	8.910	0.090	1.0	5.400	1.670
**Mg**(**NO_3_**)**_2_•6H_2_O** (**mmol**)					
**1Mg−nHA**	8.910	0.090	1.0	5.400	1.670
**3Mg−nHA**	8.730	0.270	3.0	5.400	1.670
**5Mg−nHA**	8.550	0.450	5.0	5.400	1.670
**7Mg−nHA**	8.370	0.630	7.0	5.400	1.670
**Zn**(**NO_3_**)**_2_ •6H_2_O** (**mmol**)					
**1Zn−nHA**	8.910	0.090	1.0	5.400	1.670
**3Zn−nHA**	8.730	0.270	3.0	5.400	1.670
**5Zn−nHA**	8.550	0.450	5.0	5.400	1.670
**7Zn−nHA**	8.370	0.630	7.0	5.400	1.670

**^a^** M: Cu^2+^, Mg^2+^, and Zn^2+^ ions.

## Data Availability

Data is contained within the article.

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
