# Peer review of "Morphological Changes, Antibacterial Activity, and Cytotoxicity Characterization of Hydrothermally Synthesized Metal Ions-Incorporated Nanoapatites for Biomedical Application"

_pharmaceuticals, 2022, doi:10.3390/ph15070885_

Round 1

Reviewer 1 Report

A pleasure reading your work. However, a few notes:

- The introduction section needs some work. Please consider broader works on antimicrobial activities of metal compounds (i.e. Baldelli, A., Etayash, H., Oguzlu, H., Mandal, R., Jiang, F., Hancock, R.E. and Pratap-Singh, A., 2022. Antimicrobial properties of spray-dried cellulose nanocrystals and metal oxide-based nanoparticles-in-microspheres. Chemical Engineering Journal Advances10, p.100273.)

- The conclusion and abstract sections can be improved to align the great quality of your results.

- Some mistakes in the English grammar are present.

Author Response

Detailed responses to Reviewer #1 comments:

Dear Reviewer, thanks a lot for your positive feedback concerning our work.

We have tried to take into consideration all of your valuable advices and comments and we hope that these improvements made to our article will be helpful for readers of the Pharmaceuticals journal.

Now let us proceed to your comments. All our changes made in the article are highlighted.

Reviewer #1

A pleasure reading your work. However, a few notes:

- The introduction section needs some work. Please consider broader works on antimicrobial activities of metal compounds (i.e. Baldelli, A., Etayash, H., Oguzlu, H., Mandal, R., Jiang, F., Hancock, R.E. and Pratap-Singh, A., 2022. Antimicrobial properties of spray-dried cellulose nanocrystals and metal oxide-based nanoparticles-in-microspheres. Chemical Engineering Journal Advances10, p.100273.) - The conclusion and abstract sections can be improved to align the great quality of your results.

Response: Thank you for the remark. We have improved the presentations in the Abstract, Introduction and Conclusions according to your suggestions. We hope the modification is acceptable.

- Some mistakes in the English grammar are present.

Response: I have considered the reviewers’ comment and asked a proofreader to check the revised manuscript. The grammar and style have been corrected throughout the manuscript.

Reviewer 2 Report

In this work, the authors prepared nanoapatites with different metal ions by hydrothermal method for biomedical application. However, the work still needs some modifications to meet the high quality for publication. Please see following:

1)     They just studied the effect of different metal ions, have they considered to study the effect of the contents effect of metal ions? It’s better to mention them as well.

2)     In abstract, line 26th, Please write it clearly. “High magnification electron images” Is it SEM or TEM?; line 28th to 30th, there is a grammar mistake in this sentence “The bac-teriostatic….”. Please rewrite it.

3)     In abstract, the authors sometimes used past tense, sometimes used present tense. Please keep them consistent.

4)     Why they use "nanoparticles" to serve as the keywords? It's better to use "nanomaterials" since they mentioned the length of nanopatites. If they are nanoparticles, it should be "diameter" instead of "length",

5)     Please add more information they have done in this work at the end of introduction like characterizations. Otherwise, it's too less.

6)     Please provide the y axis in FTIR and XRD Figures.

7)     In TEM, they have already provided the scale bar in the image, why they draw them again at the end of images?

8)     My suggestion is to separate Figure 3 (a) to (d) since every Figure has too many images. Or they can just put one to represent and put others into the supporting information.

9)     For Figure 4 and Figure 5, why not put (a)(b)(c) together?

10)  Please provide the purity of chemicals in part 3.1.

11)  In page 15, line 285th, "CH3COOH" the number 3 should be subscript.

12)  In FTIR part, they can cite the reference: Materials Today Physics, 2021, 21, 100502. In other part, they can cite the following references: Engineered Science, 2021, 15, 105-115, https://dx.doi.org/10.30919/es8d492; Adv. Compos. Hybrid Mater. 2020, 3, 8-15. https://doi.org/10.1007/s42114-020-00141-9; Adv. Compos. Hybrid Mater. 2018, 1, 759–765. https://doi.org/10.1007/s42114-018-0056-z.

Author Response

Detailed responses to Reviewer #3 comments:

Dear reviewer,

Thank you very much for your review work. We have tried to take into consideration all of your valuable advices and comments and we hope that these improvements made to our article will be helpful for readers of the Pharmaceuticals journal.

Reviewer #3

In this work, the authors prepared nanoapatites with different metal ions by hydrothermal method for biomedical application. However, the work still needs some modifications to meet the high quality for publication. Please see following:

  • They just studied the effect of different metal ions, have they considered to study the effect of the contents effect of metal ions? It’s better to mention them as well.

Response: Thanks for your suggestion. Based on the reviewers' suggestions, the revised manuscript addresses the issue of the effect of metal ion content to a large extent based on improvements to the presentations (red marks in revised manuscript).

2)     In abstract, line 26th, Please write it clearly. “High magnification electron images” Is it SEM or TEM?; line 28th to 30th, there is a grammar mistake in this sentence “The bac-teriostatic….”. Please rewrite it.

Response: Thanks for your suggestion. We have edited the abstract as you suggested.

  • In abstract, the authors sometimes used past tense, sometimes used present tense. Please keep them consistent.

Response: Thanks for your suggestion. We unified the tenses as you suggested.

4)     Why they use "nanoparticles" to serve as the keywords? It's better to use "nanomaterials" since they mentioned the length of nanopatites. If they are nanoparticles, it should be "diameter" instead of "length",

Response: Thanks for your suggestion. We've changed the keywords based on your suggestion.

5)     Please add more information they have done in this work at the end of introduction like characterizations. Otherwise, it's too less.

Response: Thanks for your suggestion. We followed your suggestion and added the informations on page 3 (marked in red) of the revised manuscript.

6)     Please provide the y axis in FTIR and XRD Figures.

   Response: We appreciate the efforts of the Reviewers. The y-axis has been provided in revised Figures 1 and 2.

7)     In TEM, they have already provided the scale bar in the image, why they draw them again at the end of images?

Response: Thanks for your suggestion. To prevent confusion, we have moved the provided scale bar in the revised Figure 3.

8)     My suggestion is to separate Figure 3 (a) to (d) since every Figure has too many images. Or they can just put one to represent and put others into the supporting information.

Response: Thanks for your suggestion. We modified the image arrangement in the modified Figure 3.

9)     For Figure 4 and Figure 5, why not put (a)、(b)、(c) together?

   Response: Thanks for your suggestion. Figures 4 and 5 have been modified as per your suggestion.

10)  Please provide the purity of chemicals in part 3.1.

   Response: We appreciate the efforts of the Reviewers. Purities for all chemicals have been provided in the revised Section 2.1. (Red marks)

11)  In page 15, line 285th, "CH3COOH" the number 3 should be subscript.

   Response: Thanks for checking, we have corrected it.

12)  In FTIR part, they can cite the reference: Materials Today Physics, 2021, 21, 100502.

   Response: Thanks for your suggestion. The newly added reference [48] is addressed in revised manuscript.

   In other part, they can cite the following references: Engineered Science, 2021, 15, 105-115, https://dx.doi.org/10.30919/es8d492; Adv. Compos. Hybrid Mater. 2020, 3, 8-15. https://doi.org/10.1007/s42114-020-00141-9; Adv. Compos. Hybrid Mater. 2018, 1, 759–765. https://doi.org/10.1007/s42114-018-0056-z.;

      Response: Thanks for your suggestion. The newly added references [49-51] is addressed in revised manuscript.

Reviewer 3 Report

Authors have studied the morphological changes, antibacterial activity, and cytotoxicity characterization of hydrothermally synthesized metal ions- incorporated nanoapatites for biomedical application, however, the simple XTT analysis of cell cytotoxicity shows cytotoxic effect of these nanoapatite which limits the biomedical application. Though antibacterial effects are prominent in nanoapatite and the structural characterization are complementary, yet it needs major revision in terms of discussion of the results more specifically the cell viability with reference to literature. Moreover, manuscript has a lot of ambiguous statements which are not easy to understand, therefore, it is advised to please thoroughly read the article and in light of the issues mentioned here.

1.      What do authors mean by “Line-25; These phenomena were observed in the high-magnification electron images” in abstract?

2.      Materials and method section can be after the introduction and before results and discussion.

3.      Please cross-check the method in section 3.2 Preparation of Nano-hydroxyapatite with Metal Ions (Cu2+, Mg2+, and Zn2+) 287 and clearly explain the procedure specially first two lines are vague. Moreover, why authors used different metal ion content for Cu2+ in comparison to Mg2+ and Zn2+? Any reference?

4.      Section “3.3. Characterization of Nanoparticles through Fourier Transform Infrared (FTIR) Spectroscopy, 301 X-Ray Diffraction (XRD) Analysis, and Transmission Electron Microscopy (TEM) 302 Morphological Measurements “needs to be clearly explained again. For example, “The functional groups of the nanoparticles were analyzed using the generated hy-304 droxyl groups (OH−), and the wavenumber during the formation of HA was between 3500 305 and 3600 cm−1 and used as the basis for Fourier” hard to understand.

5.      Similarly, in section 3.4., what do author mean to say by “Line -330 The bacterial suspensions were diluted to achieve an optical density of 595 nm (OD595).”?

6.      For cytocompatibility test, authors have used the extraction weight ratio (g/mL) of the sample to the medium at 1:5, wing to the hydrophilicity of the nanoapatites, can author cite some literature for this particular ratio and its relationship to the hydrophilicity of NPs? Cytotoxicity of the nanoapatite needs to be revised with reference to support “The nanoapatite groups with Cu2+ showed cytotoxicity, whereas nanoapatites with Mg2+ and Zn2+ and the 7Mg−nHA, 3Zn−nHA, 5Zn−nHA, and 7Zn−nHA groups still showed cytotoxicity.” Why does authors chose XTT cell proliferation assay kit for cell viability?

7.      Line-183-185 “The particle sizes of Cu2+,  Mg2+, and Zn2+ ions were much smaller than the particle size of Ca2+ and can thus freely enter and exit bacterial cells, disrupt cell wall synthesis, and reduce the protective effect of bacteria.” Please check the statement and provide some reference to support Cu2+ being better at antibacterial effect than Mg2+ and Zn2+.

8.      How can author explain the results “Low concentration Cu2+ has better antibacterial but more cytotoxic effect” for biomedical applications?

9.      Authors have concluded that “the Zn2+ ionic group showed the worst performance in terms of antibacterial ability and biocompatibility”. However, literature counts Zn has been used as better antibacterial agent, please check and support your findings with literature.

Please provide references to these sentences;

Line-41; Hydroxyapatite (HA) with molecular formula Ca10(PO4)6(OH)2 is a calcium phosphate-based bioceramic.

Line-42; Apatite has excellent biocompatibility and osteoconductivity, similar to inorganic compounds in human bones and teeth.

Line-53; Although apatite offers benefits as a biomaterial implant, it can affect the proliferation rate of osteoblasts and the expression of cytokine genes.

Line-57, 58; Proper antibiotic prophylaxis can reduce the risk of  postoperative wound infection, but antibiotics are becoming increasingly ineffective as drug resistance spreads globally.

Line-75, 76; Approximately 60% of Mg2+ in the human body is stored in the bones, and Mg2+ regulates osteoblast or osteoclast activities.

Line-180; The cell wall structure of E. coli is more complex than that of S. aureus,

Line-210,211 According to the standard ISO10993-5 routine, cell viability was less than 70%, that is, the extracts were toxic to cells.

Grammatical error;

Line-46, 47; HA can be synthesized through precipitation [4, 5], and with sol-gel [6, 7], hydrothermal [8-10], and microwave-assisted [11, 12] methods and HA can be extracted from biological resources [13-15].

Line-54; Moreover, apatite lacks an antibacterial mechanism.

Line-59; Hence, treating infections and preventing deaths have become challenging.

Line-165; As for the group with Zn2+, as the Zn2+ ionic addition amount increases, the bacteriostatic ability is enhanced (Figure 4c).

Line-210,211 According to the standard ISO10993-5 routine, cell viability was less than 70%, that is, the extracts were toxic to cells.

Line 241-243; “Arul et al. [56] used a microwave strategy to rapidly synthesize Mg2+-incorporated apatite nanorods; they found that the incorporation of Mg2+ ions did not change the phase, but the crystallite and particle sizes were significantly reduced by 48% and 32%, respectively. Their results”

Minor issues;
Line-356; “and then.100 μL of the sample extract” There should be no full stop “.” after then

Author Response

Detailed responses to Reviewer #2 comments:

We greatly appreciate the efforts of the Reviewer to improve the manuscript. All comments on review recommendations have been processed.

Reviewer #2

Authors have studied the morphological changes, antibacterial activity, and cytotoxicity characterization of hydrothermally synthesized metal ions- incorporated nanoapatites for biomedical application, however, the simple XTT analysis of cell cytotoxicity shows cytotoxic effect of these nanoapatite which limits the biomedical application. Though antibacterial effects are prominent in nanoapatite and the structural characterization are complementary, yet it needs major revision in terms of discussion of the results more specifically the cell viability with reference to literature. Moreover, manuscript has a lot of ambiguous statements which are not easy to understand, therefore, it is advised to please thoroughly read the article and in light of the issues mentioned here.

  1. What do authors mean by “Line-25; These phenomena were observed in the high-magnification electron images” in abstract?

Response: We thank the reviewers for their attention to this issue. We have revised the incorrect statement (red marks in Abstract section).

  1. Materials and method section can be after the introduction and before results and discussion.

   Response: Thank you for the suggestion, we have rearranged the sections by Materials and methods after the introduction.

  1. Please cross-check the method in section 3.2 Preparation of Nano-hydroxyapatite with Metal Ions (Cu2+, Mg2+, and Zn2+) 287 and clearly explain the procedure specially first two lines are vague. Moreover, why authors used different metal ion content for Cu2+ in comparison to Mg2+ and Zn2+? Any reference?

   Response: We have revised the sentence (Lines 1-6, Section 2.2. In the revised manuscript). The newly added reference [47] is also addressed.  

  1. Section “3.3. Characterization of Nanoparticles through Fourier Transform Infrared (FTIR) Spectroscopy, 301 X-Ray Diffraction (XRD) Analysis, and Transmission Electron Microscopy (TEM) 302 Morphological Measurements “needs to be clearly explained again. For example, “The functional groups of the nanoparticles were analyzed using the generated hy-304 droxyl groups (OH−), and the wavenumber during the formation of HA was between 3500 305 and 3600 cm−1 and used as the basis for Fourier” hard to understand.

Response: We appreciate the Reviewer for the suggestion. Accordingly, we have rewritten many parts of the revised Section 2.3. (Red marks)

  1. Similarly, in section 3.4., what do author mean to say by “Line -330 The bacterial suspensions were diluted to achieve an optical density of 595 nm (OD595).”?

Response: Thanks again to the reviewers for the detailed review. Therefore, we have rewritten the sentences of the revised Section 2.4. (Lines 2-4, red marks)

  1. For cytocompatibility test, authors have used the extraction weight ratio (g/mL) of the sample to the medium at 1:5, wing to the hydrophilicity of the nanoapatites, can author cite some literature for this particular ratio and its relationship to the hydrophilicity of NPs?

Response: We thank the reviewers for their suggestions. The suggested relevant reference about hydrophlicity of the nanoapatite has been added (new citation #52).

   Cytotoxicity of the nanoapatite needs to be revised with reference to support “The nanoapatite groups with Cu2+ showed cytotoxicity, whereas nanoapatites with Mg2+ and Zn2+ and the 7Mg−nHA, 3Zn−nHA, 5Zn−nHA, and 7Zn−nHA groups still showed cytotoxicity.”

Response: We thank the reviewers for their suggestions. The suggested relevant reference about cytotoxicity of the nanoapatite has been added (new citation #68.

   Why does authors chose XTT cell proliferation assay kit for cell viability?

Response: Thanks for the suggestion. The key reasons for using the XTT rather than the MTT cell proliferation assay kit to detect L929 cell viability are the rapid assay (without the solubilization step in the MTT assay), reproducible and sensitive results in a microplate.

  1. Line-183-185 “The particle sizes of Cu2+,  Mg2+, and Zn2+ ions were much smaller than the particle size of Ca2+ and can thus freely enter and exit bacterial cells, disrupt cell wall synthesis, and reduce the protective effect of bacteria.” Please check the statement and provide some reference to support Cu2+ being better at antibacterial effect than Mg2+ and Zn2+.

   Response: We appreciate the efforts of the Reviewers. We have complied with the Reviewers’ comments and made changes accordingly. The suggested relevant reference about Cu2+ being better at antibacterial effect than Mg2+ and Zn2+ has been added (new citation #65).

  1. How can author explain the results “Low concentration Cu2+ has better antibacterial but more cytotoxic effect” for biomedical applications?

      Response: Thanks for the suggestion. The suggestion has been has been disclosed in the conclusions of the revised manuscript. (Future studies should consider expanding the testing range, for example, if the 0.2% Cu2+ doped nanorods with the strongest antibacterial activity in this study were incorporated into other biocompatible medical devices, it might reduce cytotoxicity but maintain bacteriostatic properties.)

  1. Authors have concluded that “the Zn2+ ionic group showed the worst performance in terms of antibacterial ability and biocompatibility”. However, literature counts Zn has been used as better antibacterial agent, please check and support your findings with literature.

   Response: We appreciate the efforts of the Reviewers. Most of the differences regarding the performance of Zn2+ can be attributed to the concentrations studied. Accordingly, the suggested relevant reference about Cu2+ being better at antibacterial effect than Mg2+ and Zn2+ has been added (new citation #65). 

Please provide references to these sentences;

Line-41; Hydroxyapatite (HA) with molecular formula Ca10(PO4)6(OH)2 is a calcium phosphate-based bioceramic. 

Response: Thanks for your comments, we have cited reference [5] in the revised manuscript, page 2.

Line-42; Apatite has excellent biocompatibility and osteoconductivity, similar to inorganic compounds in human bones and teeth.

Response: Thanks for your comments, we have cited reference [6] in the revised manuscript, page 2.

Line-53; Although apatite offers benefits as a biomaterial implant, it can affect the proliferation rate of osteoblasts and the expression of cytokine genes.

Response: Thanks for your comments, we have cited reference [26, 27] in the revised manuscript, page 2.

Line-57, 58; Proper antibiotic prophylaxis can reduce the risk of  postoperative wound infection, but antibiotics are becoming increasingly ineffective as drug resistance spreads globally.

Response: Thanks for your comments, we have cited reference [27] in the revised manuscript, page 2.

Line-75, 76; Approximately 60% of Mg2+ in the human body is stored in the bones, and Mg2+ regulates osteoblast or osteoclast activities.

Response: Thanks for your comments, we have cited reference [41] in the revised manuscript, page 2.

Line-180; The cell wall structure of E. coli is more complex than that of S. aureus,

Response: Thanks for your comments, we have cited reference [62] in the revised manuscript, page 11.

Line-210,211 According to the standard ISO10993-5 routine, cell viability was less than 70%, that is, the extracts were toxic to cells.

Response: Thanks for your comments, we have cited reference [67] in the revised manuscript, page 12.

Grammatical error;

Line-46, 47; HA can be synthesized through precipitation [4, 5], and with sol-gel [6, 7], hydrothermal [8-10], and microwave-assisted [11, 12] methods and HA can be extracted from biological resources [13-15].

Line-54; Moreover, apatite lacks an antibacterial mechanism.

Line-59; Hence, treating infections and preventing deaths have become challenging.

Line-165; As for the group with Zn2+, as the Zn2+ ionic addition amount increases, the bacteriostatic ability is enhanced (Figure 4c).

Line-210,211 According to the standard ISO10993-5 routine, cell viability was less than 70%, that is, the extracts were toxic to cells.

Line 241-243; “Arul et al. [56] used a microwave strategy to rapidly synthesize Mg2+-incorporated apatite nanorods; they found that the incorporation of Mg2+ ions did not change the phase, but the crystallite and particle sizes were significantly reduced by 48% and 32%, respectively. Their results”

Responses about the above mentioned grammatical errors: We appreciate the efforts of the Reviewers. We have complied with the Reviewers’ comments and made changes accordingly. In addition, we have asked a proofreader to check the revised manuscript. Grammar and style have been corrected throughout the manuscript.

Minor issues;
Line-356; “and then.100 μL of the sample extract” There should be no full stop “.” after then

 Response: We sincerely appreciate the efforts of the Reviewer. The problem is solved.

Round 2

Reviewer 2 Report

The paper has been well revised. 

Reviewer 3 Report

All the comments have been addressed.